# Boosting quantum yields in two-dimensional semiconductors via proximal metal plates

Yongjun Lee[1,10], Johnathas D'arf Severo Forte[2,10], Andrey Chaves [2,3,10], Anshuman Kumar [4], Trang Thu Tran[1], Youngbum Kim[1], Shrawan Roy[1], Takashi Taniguchi [5], Kenji Watanabe [6], Alexey Chernikov [7], Joon I. Jang [8], Tony Low[9✉] & Jeongyong Kim [1✉]

Monolayer transition metal dichalcogenides (1L-TMDs) have tremendous potential as atomically thin, direct bandgap semiconductors that can be used as convenient building blocks for quantum photonic devices. However, the short exciton lifetime due to the defect traps and the strong exciton-exciton interaction in TMDs has significantly limited the efficiency of exciton emission from this class of materials. Here, we show that exciton-exciton interaction in 1L-WS$_2$ can be effectively screened using an ultra-flat Au film substrate separated by multilayers of hexagonal boron nitride. Under this geometry, induced dipolar exciton-exciton interaction becomes quadrupole-quadrupole interaction because of effective image dipoles formed within the metal. The suppressed exciton-exciton interaction leads to a significantly improved quantum yield by an order of magnitude, which is also accompanied by a reduction in the exciton-exciton annihilation (EEA) rate, as confirmed by time-resolved optical measurements. A theoretical model accounting for the screening of the dipole-dipole interaction is in a good agreement with the dependence of EEA on exciton densities. Our results suggest that fundamental EEA processes in the TMD can be engineered through proximal metallic screening, which represents a practical approach towards high-efficiency 2D light emitters.

[1] Department of Energy Science, Sungkyunkwan University, Suwon 16419, Republic of Korea. [2] Departamento de Física, Universidade Federal do Ceará, Campus do Pici, 60455-900 Fortaleza, Ceará, Brazil. [3] Department of Physics, University of Antwerp, Groenenborgerlaan 171, B-2020 Antwerpen, Belgium. [4] Physics Department, Indian Institute of Technology Bombay, Mumbai 400076, India. [5] International Center for Materials Nanoarchitectonics, National Institute for Materials Science, 1-1 Namiki, Tsukuba 305-0044, Japan. [6] Research Center for Functional Materials, National Institute for Materials Science, 1-1 Namiki, Tsukuba 305-0044, Japan. [7] Dresden Integrated Center for Applied Physics and Photonic Materials (IAPP) and Würzburg-Dresden Cluster of Excellence ct.qmat, Technische Universität Dresden, 01062 Dresden, Germany. [8] Department of Physics, Sogang University, Seoul 04107, Republic of Korea. [9] Department of Electrical & Computer Engineering, University of Minnesota, Minneapolis, MN 55455, USA. [10] These authors contributed equally: Yongjun Lee, Johnathas D'arf Severo Forte, Andrey Chaves. ✉email: tlow@umn.edu; j.kim@skku.edu

Transition metal dichalcogenides (TMDs) are layered materials where the forces between its constituent layers are mediated by weak van der Waals interaction. In particular, the 1H phase of TMD monolayers with $MX_2$ (M = Mo, W and X = S, Se) stoichiometry has a direct bandgap at the high symmetry K (K′) points of the Brillouin zone[1,2]. Due to strong two-dimensional (2D) confinement, Coulomb bound electron-hole pairs, commonly known as excitons, can be formed upon optical excitation within the light cone at the K (K') points, where they would subsequently undergo radiative recombination to yield photoluminescence (PL)[3]. However, the quantum yield (QY) of monolayer (1 L)-$MX_2$ is fundamentally limited by various nonradiative decay pathways. This includes the conversion of bright excitonic states into spin- and valley-forbidden states (dark states)[3,4]. Numerous efforts have been recently made to improve the QY by investigating and/or even controlling the limiting mechanisms. It was found that as-exfoliated semiconducting TMDs, such as 1L-$WS_2$ and 1L-$MoS_2$, are inherently n-type, which have excess free carriers and thus promoting the formation of trions and their nonradiative recombination[5–7]. In addition, the presence of a large number of defects in 1L-TMDs (especially chalcogen vacancies, $\sim10^{12}$ cm$^{-2}$) can serve as nonradiative recombination centers leading to low QYs[8–10]. Various methods such as defect passivation or charge transfer were proposed to improve the QY[7,10–14]. For example, a near-unity QY in sulfur-based 1L-$MS_2$ has been reported through passivation of defect-mediated nonradiative recombination by super-acid treatment[12] and atomic healing of sulfur vacancies were also observed by scanning transmission electron microscopy[10]. The current understanding is that the compensation of n-doping, such as via chemical treatment or by electrostatic doping, restores the predominance of neutral excitons, and therefore, enhance the PL efficiency of the 1L-TMDs[12–14].

However, even in the ideal pristine 2D $MX_2$, the QY can still be quenched when the exciton-exciton interaction dominates to cause strong exciton Auger recombination, so-called exciton-exciton annihilation (EEA)[15–19]. This density-dependent fundamental process occurs at high exciton densities under strong light illumination. Most notably, experimentally reported EEA rate constants revealed a very large variation across different device configurations with values ranging from a few $10^{-1}$ cm$^2$/s to $10^{-3}$ cm$^2$/s for 1L-$WS_2$. These reported EEA rate constants are also orders of magnitude higher than those typically observed in semiconducting quantum-well systems[20]. Since electroluminescence (EL) in 2D TMDs arises as a consequence of radiative excitonic transition[21], EEA practically determines the quantum efficiency of a light-emitting device at the exciton density under typical brightness[22,23]. Without question, the EEA process represents one of the key technological bottlenecks for achieving energy-efficient and bright EL devices even in otherwise pure, high-quality samples[8]. Despite experimental reports that the use of high-index substrates[24] or hBN encapsulation[25,26] resulted in largely reduced EEA rates in 1L-TMDs accompanied by the relative increase of PL, its physical origin was not well understood. Active suppressions of EEA were reported through the intentional formation of defects[8,27], or by applying the strain[28]. Fundamental understanding of EEA, and strategies in alleviating EEA and the accompanying QY drooping at high exciton densities are of utmost importance to high-efficiency light-emitting devices.

Although an exciton in the 1 s state is charge neutral and has no higher multipole moments, it can assume an induced dipole moment by the presence of other exciton. As schematically depicted in Fig. 1a, the dipolar interaction between the electron-hole pairs that form the excitons and its Coulombic fields can be strongly influenced by the presence of a proximal metallic plate because the image dipoles in the metal renders exciton-exciton interaction essentially quadrupolar. In this work, we provide the first systematic study of EEA and QY enhancement in this device geometry using ultra-flat Au substrates with atomically controlled hBN spacers. Our simple approach of controlling the exciton interaction via proximal metal plates provides an efficient and feasible route towards practical applications using highly efficient light-emitting devices derived from 1L-TMDs.

## Results and discussion

**Substrate-dependent PL intensity at low-density regime.** Figure 1a illustrates a schematic of the sample configuration, with 1L-$WS_2$ prepared on an ultra-flat Au surface separated by a multilayered hBN spacer to serve as an insulating barrier, compared with 1L-$WS_2$ directly on $SiO_2$. Coulombic fields between two excitons would be substantially suppressed by metallic screening due to the presence of the Au substrate. Our goal is to investigate the anticipated reduction in Auger scattering between excitons, or EEA, due to such screening. In Fig. 1b, c, the atomic force microscopy (AFM) image and the PL intensity map of the actual sample are shown, respectively, where the thickness of the hBN layer on the Au surface was measured to vary in the range of 7–36 nm. We used the low laser power of 2 nW to assess the QY without the effect of EEA. The PL spectra of 1L-$WS_2$ obtained on the $SiO_2$/Si substrate and on Au at several hBN thicknesses are displayed in Fig. 1d. As shown in Fig. 1c, d, we observed that the PL intensity is highly dependent on the hBN thickness, where the corresponding PL intensities are summarized in Fig. 1e as a function of hBN thickness. Multiple samples were investigated, and the measured data obtained from the different 1L-$WS_2$ flakes are denoted by different symbols. We found that the PL intensities of 1L-$WS_2$ samples showed a consistent general trend, namely, a gradual increase of PL with the hBN thickness up to an optimum thickness of 33 nm, followed by a rapid decrease beyond this value.

Optical interference has to be accounted for in order to understand the origin of the measured changes in the PL intensity. The detected PL is proportional to both the optical absorption and the exciton emission yields of 1L-$MX_2$, hence it is critically affected by optical interference due to the multi-layered substrate[29,30]. To properly account for the optical interference effect, we modeled the optical enhancement factor (EF) as a function of hBN thickness on the Au substrate relative to the control configuration of air-suspended 1L-$WS_2$ using the Fresnel equation under normal incidence[29] (Details for the estimation of the EF are described in the Supplementary Information). The estimated EF is overlaid (red line) with the measured PL-intensity data in Fig. 1e. The calculated EF corroborates very well with the measured PL intensity, indicating that the optical interference effect dominates the observed PL dependence on the hBN thickness. Differential reflection spectra of 1L-$WS_2$ also confirmed stronger absorption at thicker hBN on the Au substrate and we showed that the effect of the hBN thickness on the measured PL intensity can be nearly flattened by encapsulating 1L-$WS_2$ with thick (~50 nm) hBN (Supplementary Figs. 2 and 3 in the Supplementary Information).

For direct comparison, we measured the PL intensity of 1L-$WS_2$ on the $SiO_2$/Si substrate, which is denoted by the star on the left vertical axis in Fig. 1e. At first glance, it seems that the relative PL intensity from 1L-$WS_2$ on 7 nm hBN/Au is 2.5 times higher than that from 1L-$WS_2$ directly on the $SiO_2$/Si substrate (Fig. 1d, e). However, we confirmed that the actual QY of the former is about five times larger since the $SiO_2$/Si substrate gains the PL brightness via the EF by more constructive inference of multiple internal reflections, as denoted by the star on the right vertical

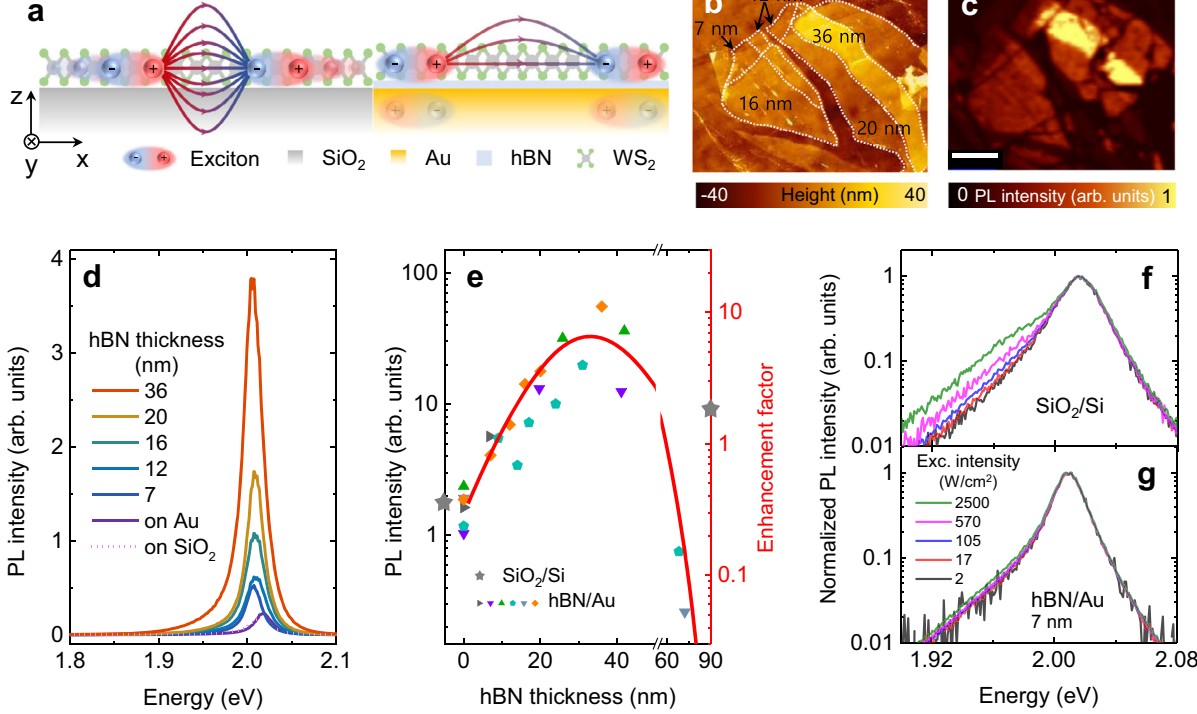

**Fig. 1 Impact of the hBN/Au substrate on the PL intensity. a** Schematic of exciton-exciton interaction suppressed by the hBN/Au substrate. Shaded arrows represent exciton-exciton interaction. Diluted field lines schematically indicate the reduced interaction by the effect of image dipoles. **b** AFM height and **c** PL intensity maps of 1L-WS$_2$ on the hBN/Au substrate. The scale bar in **c** corresponds to 7 μm. **d** Measured PL spectra of 1L-WS$_2$ on various thicknesses of hBN on Au. **e** PL intensities of six 1L-WS$_2$ samples (same color and shape of dots represent the data sets obtained from the same 1L-WS$_2$ flakes) as a function of hBN thickness on the Au surface at the excitation intensity of 2.1 W/cm$^2$. The red curve is the calculated enhancement factor (EF) due to the optical interference of the multilayered substrate relative to the suspended 1L-WS$_2$. The measured PL intensity and calculated EF of 1L-WS$_2$ on SiO$_2$/Si are indicated by the stars on the left and right Y-axes for comparison. Normalized PL spectra obtained from **f** 1L-WS$_2$ on SiO$_2$/Si and **g**. hBN (7 nm)/Au surfaces, respectively, under various excitation intensities. Above 100 W/cm$^2$, a trion peak at the low-energy shoulder near 1.98 eV appeared.

axis in Fig. 1e. This clearly suggests that the proximity to the Au surface increased the QY of 1L-WS$_2$. As a possible explanation, the proximity to Au could have resulted in depletion of the excess electrons of 1L-WS$_2$ due to the lower work function of Au than 1L-WS$_2$[31], and compensating hole doping has been shown to enhance the QY of n-type 1L-TMDs[11,14]. Under low photo-excitation intensities, the PL spectra of 1L-WS$_2$ are understood to be dominated by neutral exciton emissions. However, with increasing the excitation intensity, prior studies have found the emergence of the trion emission peak due to electrons generated from photoionized shallow donors[6,32]. In Fig. 1f, g, we show the normalized PL spectra observed from 1L-WS$_2$ on SiO$_2$/Si and hBN/Au substrates under various excitation intensities, respectively. With increasing the photo-excitation intensity, a low-energy shoulder in the PL spectra distinctively grew for the sample on the SiO$_2$ substrate, indicative of radiative recombination of trions. On the other hand, no additional low-energy shoulder in the PL spectra was visible for the sample on the hBN/Au substrate, which is consistent with our hypothesis of electrostatic depletion of electrons in 1L-WS$_2$ due to the higher conduction band edge of 1L-WS$_2$ (-4 eV) than the Fermi level of Au (-5.2 eV)[33,34].

**Substrate-dependent PL lifetime at the low-density regime.** The proximal Au surface also showed a notable impact on the lifetime of photoexcited excitons in 1L-WS$_2$. We first considered the low photo-excitation regime where EEA can be neglected. Figure 2a shows the PL intensity map of 1L-WS$_2$ flakes prepared on the SiO$_2$/Si substrate, of which some parts are sitting on the hBN

spacer of 27 nm in thickness (highlighted by the white dashed line in Fig. 2a). As shown in the PL intensity map (Fig. 2a) and the representative PL spectra obtained from 1L-WS$_2$ on SiO$_2$/Si and hBN/SiO$_2$/Si (Fig. 2b), the PL intensity from 1L-WS$_2$ on hBN/SiO$_2$/Si was lower than its counterpart on SiO$_2$/Si, whose difference can be accounted for by the optical interference effect (See the Supplementary Information). In 2D research community, hBN is frequently used as substrate and encapsulation layers for TMDs, since it preserves the pristine quality of TMDs and has the effect of refining the spectral properties such as narrowing of the spectral linewidth[25,26,35,36]. However, the role of hBN on the PL lifetime is rarely discussed. It was recently argued that hBN encapsulation alleviates EEA but also leads to an overall reduction in the PL lifetime[25]. A separate study also reported the reduction of the PL lifetime in hBN-encapsulated 1L-WS$_2$ and attributed it to enhanced exciton diffusion towards non-radiative traps[26].

We measured the PL lifetimes of 1L-WS$_2$ on hBN/SiO$_2$/Si and SiO$_2$/Si by time-resolved photoluminescence (TRPL), and their representative PL decay curves are shown in Fig. 2c. The extracted PL lifetimes were found to be 940 ± 65 ps and 143 ± 38 ps for the SiO$_2$/Si and hBN/SiO$_2$/Si substrates, respectively, as shown in Fig. 2e. Due to the very low pump fluence of 2.6 nJ/cm$^2$ (corresponding to an initial exciton density of $3.6 \times 10^8$ cm$^{-2}$), EEA can be ignored and the PL decay curves indeed exhibit a simple trend of a single exponential decay. In this regime, the PL lifetime of 1L-TMDs should be mainly determined by the rate of nonradiative capture by lattice defects[9]. The reduced PL lifetime of 1L-WS$_2$ on hBN/SiO$_2$/Si is consistent with the previous studies[25,26].

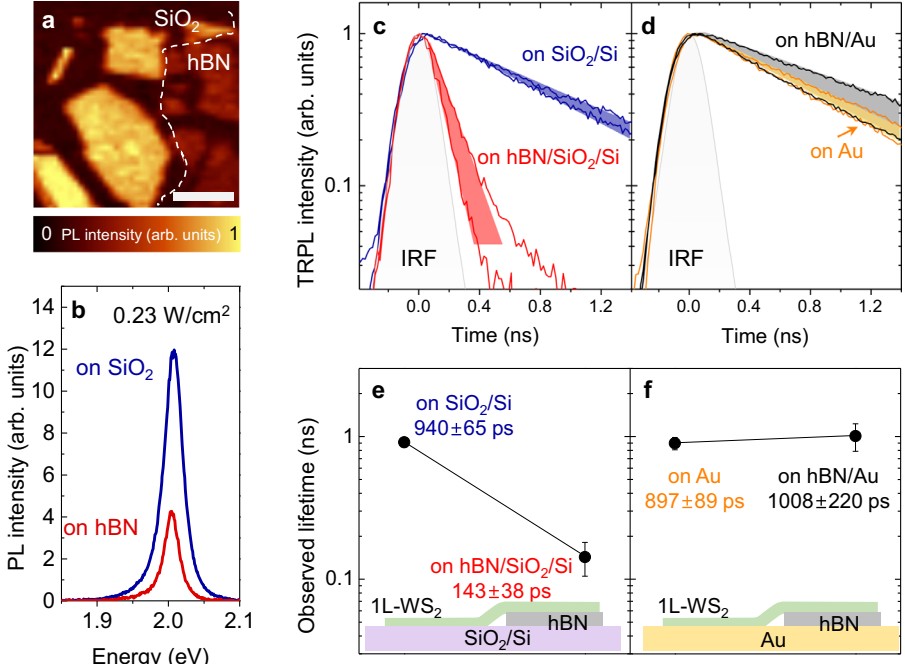

**Fig. 2 Excitonic PL at the low-density regime. a** PL intensity image and **b** PL spectrum of 1L-WS$_2$ on SiO$_2$ (dark blue) out of the white dashed region in **a** and that for 1L-WS$_2$ on hBN/SiO$_2$ (dark red) from the dashed area in **a**, respectively. The scale bar in **a** corresponds to 4 μm. Normalized TRPL data obtained at an exciton density below 5 × 10$^8$ cm$^{-2}$, free from EEA, from **c** 1L-WS$_2$ on SiO$_2$ (dark blue) and hBN/SiO$_2$ (red) and **d** 1L-WS$_2$ on Au (orange) and hBN/Au (black), respectively. Corresponding PL lifetimes are shown with the values in **e** and **f** with the same colors of the substrate conditions in **c** and **d**. The schematic structures of the samples are displayed, in which the solid lines are the guides to the eye. The colored shades in **c** and **d** indicate the range of the data trace, and the error bars in **e** and **f** indicate the standard deviation, observed from other samples with the same geometries.

Then, we examined the PL lifetimes for Au substrates. Figure 2d, f show the PL decay curves and the average PL lifetimes obtained from 1L-WS$_2$ on hBN/Au, and directly on Au. The PL lifetimes of 1L-WS$_2$ prepared on hBN/Au and Au exhibit average lifetimes of 1008 ± 220 ps and 897 ± 89 ps, respectively, which are comparable to the values of 1L-WS$_2$ prepared on the SiO$_2$/Si substrate. We observed similar trends for all the thicknesses of hBN below 20 nm, including the case for no hBN layer on the Au substrate. To the best of our knowledge, this is the first observation of the PL lifetime increase in 1L-TMDs on hBN due to the proximal Au surface. Our observation here reconciles with the concurrent enhancement of the PL efficiency due to electron depletion discussed previously in Fig. 1f, g and similarly to the PL enhancement in n-type 1L-MoS$_2$ and 1L-WS$_2$ with p-doping and defect passivation[12–14].

**PL lifetime vs. exciton density and hBN thickness.** We now discuss the effect of the Au substrate on the exciton dynamics in 1L-WS$_2$ at high exciton densities, where exciton-exciton interaction cannot be neglected. We obtained the PL transients for different exciton densities by varying pump fluences (labeled with colors shown in the legend) for our samples with hBN thicknesses of 36, 16, and 7 nm between 1L-WS$_2$ and Au. The corresponding normalized time traces are displayed in Fig. 3b–d. The normalized PL transients of 1L-WS$_2$ on the SiO$_2$ substrate are included for comparison in Fig. 3a. As the density of photoexcited excitons in 1L-WS$_2$ increases, the PL decay time gradually decreases, and such density-dependent exciton decay is characteristics of EEA in 1L-TMDs[15–18].

While Fig. 3b-d clearly show that the PL lifetimes decrease with increasing exciton densities on hBN/Au substrates, the overall effect is weaker for thinner hBN layers. For example, the short component of the PL lifetime of 1L-WS$_2$ on the 7-nm-thick hBN on the Au surface decreased from 919 ps at 1.24 × 10$^8$ cm$^{-2}$ to

only 376 ps at 1.44×10$^{11}$ cm$^{-2}$, which is a much smaller reduction (by about factor of 3) as compared to that of the sample on the SiO$_2$ substrate, where it decreases from 890 ps at 8×10$^8$ cm$^{-2}$ to 122 ps at 1.95×10$^{11}$ cm$^{-2}$. To quantify the density-dependent PL decay rate, we used the rate equation to fit the PL decay curves[16,18]:

$$\frac{dn_{ex}}{dt} = -\frac{n_{ex}}{\tau_0} - \gamma n_{ex}^2,  \quad (1)$$

where $n_{ex}$ is the exciton density, $\tau_0$ is the measured PL lifetime at a low exciton density, and $\gamma$ is the EEA rate constant. The solution to Eq. (1) can be simplified to $n_{ex}^{-1}(t) = \gamma \tau_0 \exp(t/\tau_0)$ [26], where the coefficient $\gamma$ is extracted from the slope of $n_{ex}^{-1}(t)\tau_0^{-1}$ vs $\exp(t/\tau_0)$ as shown in Fig. 3e (Justification of this approach is provided in the Supplementary Information). Figure 3f shows the plot of the estimated $\gamma$ value vs. the hBN thickness. Note that $\gamma$ of 1L-WS$_2$ on SiO$_2$ of $(1.26 \pm 0.40) \times 10^{-1}$ cm$^2$ s$^{-1}$ is consistent with previous reports on 1L-WS$_2$[16,18,37]. Such strong exciton interaction will critically degrade the QY of 1L-TMDs at high exciton densities. The estimated $\gamma$ value for 7-nm-thick hBN on the Au surface, however, was only $(1.78 \pm 0.94) \times 10^{-2}$ cm$^2$ s$^{-1}$, an order of magnitude lower than $\gamma$ of that prepared on the SiO$_2$ substrate. The dramatic improvement in the exciton lifetime at high exciton densities strongly suggests that the Au surface is very effective in screening the exciton-exciton interaction, rendering EEA much less effective. Peculiarly, directly on the Au surface where the metal screening should be maximized, $\gamma$ was measured to be $(5.43 \pm 0.61) \times 10^{-2}$ cm$^2$ s$^{-1}$, which is larger than those on the hBN layers on Au. We believe that the direct contact of 1L-WS$_2$ to the Au surface may have affected the measurement of $\gamma$ through the possible charge inhomogeneity of the Au surface and the expedited exciton decay due to energy or charge transfer. (We ignore hereafter the $\gamma$ value for direct contact with Au in discussing the trend of $\gamma$ with the thickness of hBN.) Regardless,

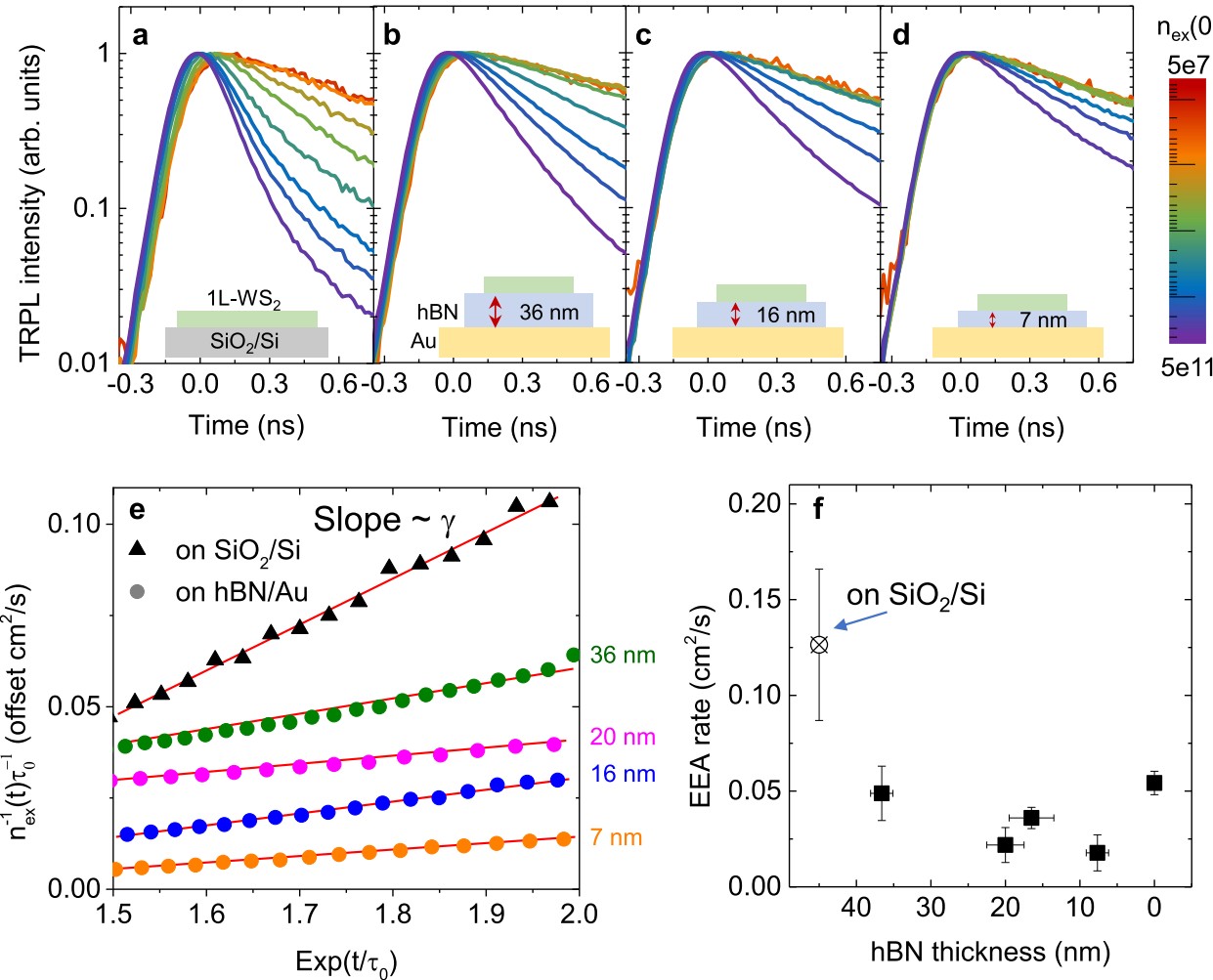

**Fig. 3 Exciton dynamics vs. hBN thickness. a–d.** Series of the normalized TRPL decay curves observed from 1L-WS$_2$ on several substrate geometries under various excitation levels (initial exciton densities). The corresponding substrate geometries are shown in the insets. The color codes in the right of **d** represent the initial exciton densities. **e** Plot of $n_{ex}^{-1}(t)\tau_0^{-1}$ vs. $\exp(t/\tau_0)$ for 1L-WS$_2$ on SiO$_2$/Si and hBN on Au for several thicknesses of hBN (shown on the right axis). The data are vertically offset for comparison. The EEA rate is extracted from the slope of linear fitting as shown by red solid lines. **f** Estimated EEA rate as a function of hBN thickness on Au (black squares) and directly on SiO$_2$/Si (crossed circle). The error bars indicate the standard deviations of the hBN thickness measurement and the EEA rates estimated at different exciton densities.

we emphasize the robust general trend of decreasing $\gamma$ with the increased proximity of 1L-WS$_2$ to the metal substrate. The PL measurement of hBN-encapsulated 1L-WS$_2$ on Au also confirmed that the effect of metal proximity suppressing EEA is higher at thinner hBN on Au (Supplementary Fig. 4).

**Enhanced absolute QY by suppressed EEA.** To quantitatively assess the effect of metal screening on exciton emission of 1L-WS$_2$ at high exciton densities, we measured the absolute QY (a-QY), the ratio of the emitted photon number to the absorbed photon number, of 1L-WS$_2$ prepared on the Au surface with the 7-nm-thick hBN spacer as a function of generation rate (Fig. 4a) (Details for the a-QY measurement of 1L-WS$_2$ and the estimation of the generation rate from the excitation intensity are described in the Supplementary Information). The a-QY of 1L-WS$_2$ on the SiO$_2$ substrate was also measured under the same experimental conditions. The plots of the a-QY vs. the generation rate ($G$) were fitted (solid curves) with the modified rate equation in Eq. (1), by adding generation term, $G$ for continuous-wave (CW) excitation with the EEA rate constants as fitting constants (fitting process and result are discussed in detail later). The a-QYs of 1L-WS$_2$ either on hBN/Au or SiO$_2$ substrates exhibit a stiff droop,

regardless of the substrate configuration, as the excitation or exciton density increases, which is typical of EEA. However, the onset of the QY droop for the two cases is quite different; more specifically, the QY droop in the hBN/Au substrate occurred at much higher excitation levels, herein found to be more than an order of magnitude higher, strongly supporting the conclusion that EEA is effectively suppressed in this structure. For example, we provide the PL spectra of 1L-WS$_2$ (adjusted with the substrate effect on absorption and emission to show the relative QYs) on hBN/Au (black) and on SiO$_2$ (orange) at $G$ of ~6.8 × 10$^{16}$ cm$^{-2}$s$^{-1}$ and ~7.5 × 10$^{20}$ cm$^{-2}$s$^{-1}$ (indicated with the light blue arrows in Fig. 4a) in Fig. 4b, c, where 3 times and 18 times enhancements in the a-QY were observed, respectively, clearly demonstrating the role of the hBN/Au substrate in effectively screening the Coulomb interaction between excitons, thereby suppressing EEA.

**Theoretical model and analysis.** In what follows, we present a theoretical model to understand the effects of the proximal metal gate on EEA and the QY. From classical electrostatics, the introduction of a perfect conductor (i.e., a metal) in a system containing charges screens the electrostatic fields. The reason for

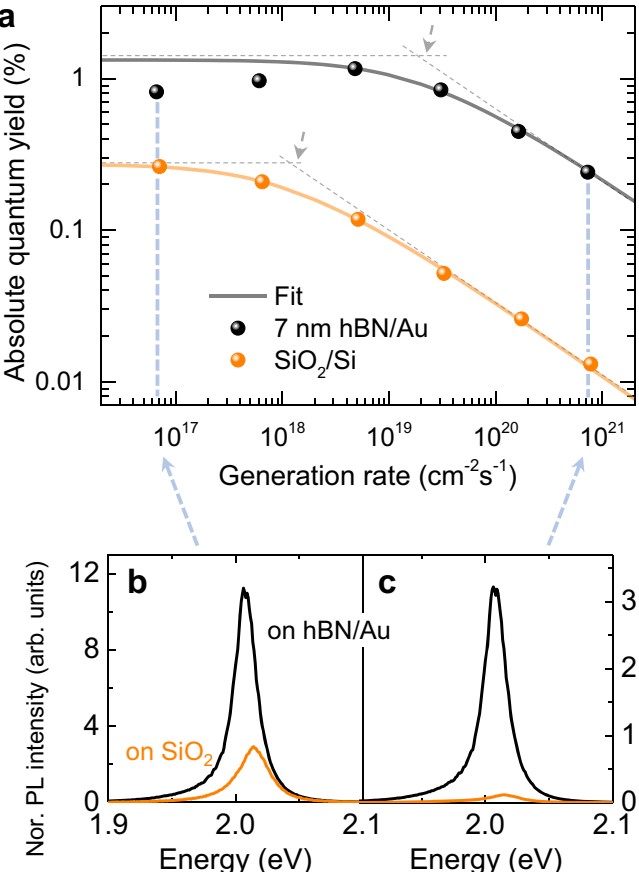

**Fig. 4 Enhanced QY by suppressed EEA. a** a-QY of 1L-WS$_2$ on hBN (7 nm)/Au (black dots) and SiO$_2$/Si (orange dots) under the generation rate (G) varying from $2 \times 10^{16}$ cm$^{-2}$s$^{-1}$ to $2 \times 10^{21}$ cm$^{-2}$s$^{-1}$ with the substrate interference effects properly taken into account. The solid curves with each color code are fits to the experimental QY calculated with Eq. (5). The dotted lines are guides for eyes and mark the points where the QY droop starts (grey arrows). **b** and **c** PL spectra of 1L-WS$_2$ on hBN (7 nm)/ Au (black) and SiO$_2$/Si (orange) at the G value indicated by light blue arrows in **a**. The PL spectra are adjusted with the substrate interference effect to show the relative a-QYs.

this is that the metal effectively creates mirror charges opposite to the original ones, transforming the charges and dipoles in the original system effectively into dipoles and quadrupoles, respectively, which renders the electric fields to be more screened and short-ranged[38]. In the case investigated here, where charges are in a monolayer semiconductor separated from the metal by an hBN spacer, a treatment involving image charges becomes less clear and the screened interaction potentials are more conveniently obtained by solving the Poisson equation, taking account of the various numbers of hBN layers on Au. We consider a system consisting of four regions, each with a distinct dielectric constant, representing the metal (Au), multilayer hBN, 1L-WS$_2$ and vacuum. The solution of the electrostatic potential due to an electron at the 1L-WS$_2$ region is:

$$\Phi(\rho, z) = \frac{e}{4\pi\varepsilon_0} \int_0^\infty dk \frac{J_0(k\rho)}{\varepsilon(k)}, \qquad (2)$$

where $J_0(k\rho)$ is the zeroth-order Bessel function of the first kind and $\varepsilon(k) = \varepsilon_{WS_2} / [A(k)e^{kz} + B(k)e^{-kz} + 1]$ is the static dielectric function in reciprocal space. $A(k)$ and $B(k)$ are coefficients to be determined through the application of the electrostatic boundary conditions at the interfaces between vacuum, 1L-WS$_2$, hBN and

Au, and were calculated using a transfer matrix scheme proposed in ref. [39]. A similar approach has been proposed also in ref. [40]. The only parameters needed for such calculation are the thickness and dielectric constants of 1L-WS$_2$ and hBN layers, which can be found in ref. [39]. The metal is accounted for by setting its dielectric constant to infinite as a boundary condition at the interface between hBN and Au. Note that the $A(k)$ and $B(k)$ coefficients depend on the size of the multilayer h-BN region, since its variation implies the changes in the h-BN/WS$_2$ and WS$_2$/vacuum interface positions. The electrostatic interaction between the source electron and a hole in the same region ($z = 0$) at a distance $\rho$ from the origin is thus $V_{eh}(\rho) = e\Phi(\rho)$. It can be straightforwardly demonstrated that the electrostatic potential in Eq. (2), with the effective $\varepsilon(k)$ calculated as previously described[39,40] exhibits all the expected limits: it converges (i) to the Rytova-Keldysh form[41,42] (i.e. with $\varepsilon(k)$ linear in $k$) in the limit of infinitely thick hBN spacer (i.e. when the screening due to the metal is negligible), (ii) to the Coulomb interaction potential in the presence of image charges, previously used in calculations for e.g. plasmons in graphene[43], in the limit of an infinitely thin WS$_2$ slab, and (iii) to the Coulomb interaction potential (i.e. with constant $\varepsilon(k) = \varepsilon_{WS2}$) in the limit of an infinitely thick WS$_2$ slab.

To calculate the electrostatic potential between two excitons, we first sum the contributions of all pairs of charges

$$V_{XX}(\rho_{ee}, \rho_{hh}, \rho_{12}, \rho_{21}) = e[\Phi(\rho_{ee}) + \Phi(\rho_{hh}) - \Phi(\rho_{12}) - \Phi(\rho_{21})], \qquad (3)$$

where the arguments represent the electron-electron, hole-hole, and electron-hole distances, respectively. Note that interactions between charges with the same (opposite) sign are positive (negative) due to the repulsive (attractive) force between them.

From here onward, we follow two approaches to investigate the effect of Au and hBN on EEA. In the first one, we model EEA as a nonradiative scattering problem between two excitons. Several pathways of Auger processes can lead to nonradiative exciton-exciton scattering: for example, one can consider an Auger process where the interaction between electrons in the conduction band of two different excitons lead to a final state where one of the excitons recombines, while its energy is used to ionize the other exciton, resulting in a hot unbound electron-hole pair in the final state. Other possibilities involve interactions between electrons in conduction and valence bands, and/or phonons[44,45]. We simplify our discussion by assuming, as an approximation, only the former case of scattering, as illustrated in Fig. 5a, which we refer to as the direct Auger process. Other processes are expected either to behave similarly as we increase the hBN thickness, or to exhibit negligible contributions. We point out that, despite the fact that we have an initial state composed by two excitons and a final state with a single electron-hole pair in our model, recombination of the first exciton still preserves charge and spin in the overall process. In what follows, as well as in the Supplementary Information, we will use similar notation as in refs. [19,44,45], where more details on the theoretical model for such direct Auger process can be found.

According to Fermi's golden rule, the scattering time is inversely proportional to the scattering probability which, for the direct Auger process considered here, is given by

$$\frac{1}{\tau_{EEA}} = \frac{2\pi}{\hbar} \sum_{\mathbf{K}_1, \mathbf{K}_2} |\mathbf{M}_{dir}(\mathbf{K}_1, \mathbf{K}_2, \mathbf{K}_f, \mathbf{k}_f)|^2 \delta \left[ E_g - 2E_B + E_K(\mathbf{K}_1) \right. $$
$$\left. + E_K(\mathbf{K}_2) - E_K(\mathbf{K}_f) - \frac{2\hbar^2 k_f^2}{M} \right] f(\mathbf{K}_1) f(\mathbf{K}_2), \qquad (4)$$

where $\tau_{EEA}$ is the Auger lifetime of excitons, $f(K)$ is the Boltzmann distribution function for classical exciton gas[19], $E_K(\mathbf{K}) = \frac{\hbar K^2}{2M}$ is

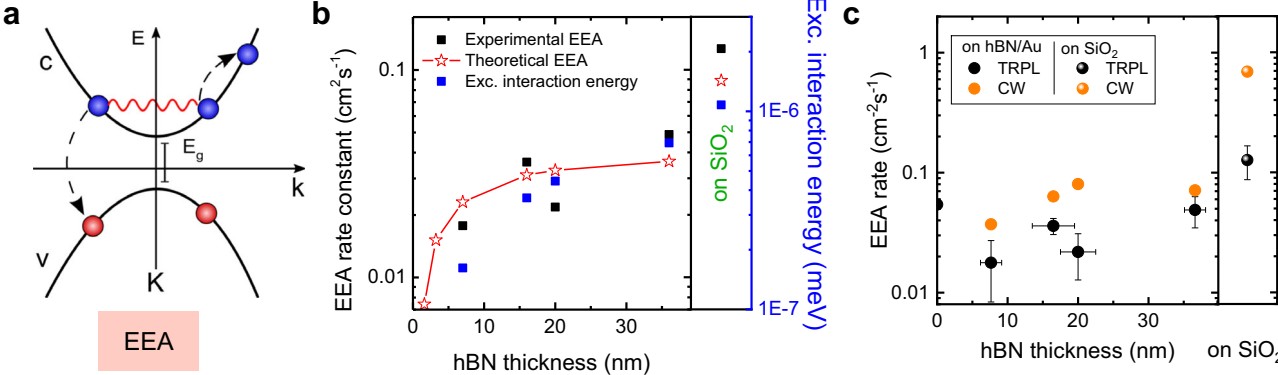

**Fig. 5 Theoretical model. a** Schematic of the nonradiative Auger scattering mechanism considered in the calculations. For this direct process, electrons from two different excitons interact via the screened potential obtained from Eq. (3). This induces the recombination of one exciton, whose energy and momentum are transferred to the other exciton, thus yielding a final state composed by a single unbound electron-hole pair. **b** Log-log plot of the measured EEA rate constant as a function of hBN thickness on Au (black squares from Fig. 3f), along with the direct Auger scattering rate constants (red stars) fitted by Eq. (4) with a proper scaling factor, and the average energy of exciton-exciton pairs at the exciton density of $1 \times 10^{11}$ cm$^{-2}$ (blue squares). A comparison between the trends in these results as a function of hBN thickness, as well as for 1L-WS$_2$ on SiO$_2$, (see symbols in the left of the panel) shows a reasonably good consistency between the measured EEA rate constants, the theoretically estimated nonradiative scattering rate constants and the exciton-exciton interaction energies. **c** Comparison of the EEA rates calculated from the TRPL experiments (the same data set as in Fig. 3f) and from the CW measurements of a-QYs, showing the same trend of increasing the EEA rate with increasing the hBN thickness on Au and highest on SiO$_2$.

the kinetic energy of the exciton, $M$ is the effective exciton mass, $E_g$ is the quasi-particle gap, $E_b$ is the exciton binding energy, and the sum is made over different values of initial momenta $\mathbf{K}_1$ and $\mathbf{K}_2$ of the initial pair of excitons. While the Dirac delta function in Eq. (4) accounts for the conservation of energy, momentum conservation is imposed in the scattering matrix term $\mathbf{M}_{dir}$, whose calculation is demonstrated in detail in the Supplementary Information. The scattering matrix term encodes the electron-electron interaction potential and exciton envelope wave functions, which are both affected by the metal screening, thus rendering $\tau_{EEA}$ effectively dependent on the hBN thickness, i.e. on the proximity to the Au substrate.

As an alternative approach to understand the observed reduction in the EEA rates as the 1L-WS$_2$ is made closer to the Au substrate, we investigate how screening due to the metal affects exciton-exciton pair interactions. In order to do so, we numerically solve the Schrödinger equation, within the Wannier-Mott model, for the two-exciton system, whose screened interaction potential is given by Eq. (3). Details of this calculation are provided in the Supplementary Information. As an approximation, we consider pairs of excitons separated by a distance R given, on average, by the inverse of the square root of the exciton density. Results are shown as the blue squares in Fig. 5b, where the dependence of the exciton-exciton interaction energy on the hBN thickness is shown together with the experimental EEA rate constants (black squares) and the theoretically estimated EEA rate constants (red stars) considering the direct Auger rate calculated by Eq. (4), for the sake of trend comparison.

Indeed, the nonradiative Auger scattering rate is indicated to be significantly reduced by the proximity to the metal, which corroborates with our experimental observation as shown in Fig. 5b. These results also indicate that 1L-hBN layer on Au could provide a much stronger screening than $N = 21$ (smallest hBN thickness in our measurement), suggesting the efficacy to suppress EEA and boost the QY with our scheme by further reducing the thickness of hBN. As shown in Fig. 5b, the measured EEA rate constants at various hBN thicknesses on Au and on SiO$_2$ substrates via the TRPL experiment follow the same trend as the theoretical ones via direct Auger scattering calculated with Eq. (4), with a single scaling factor, as well as the exciton-exciton interaction energy (see the right blue vertical axis) calculated at the exciton density of $1 \times 10^{11}$ cm$^{-2}$.

Furthermore, we related the a-QYs measured by CW laser excitation to the EEA rate by deriving an expression for the a-QYs from the rate equation[46]:

$$QY = QY_0 \left( 2 \frac{\sqrt{1 + \frac{G}{G_0}} - 1}{\frac{G}{G_0}} \right), \tag{5}$$

where $QY_0$ is the QY at the low excitation intensity (non-EEA regime) and $G_0 = 1/4\gamma\tau_0^2$. In Fig. 4a, we showed the results of fitting the QY versus $G$ data to Eq. (5) with the EEA rate constants as fitting constants for hBN thickness of 7 nm on Au and for SiO$_2$/Si substrate. The EEA rate constants determined from fitting the a-QY measurement obtained by CW laser excitation are somewhat higher than those determined via time-resolved measurements that used pulsed excitation. This discrepancy was also observed previously[26] and could originate from transient reduction of the exciton density due to diffusion of exciton at the central part of laser excitation in the TRPL measurement. However, the qualitative trend of increasing EEA rates with hBN thickness are observed in both CW and pulsed excitations, as shown in Fig. 5c.

In summary, we have demonstrated that exciton-exciton interaction and annihilation in 1L-WS$_2$ can be effectively screened by using hBN/Au substrates, improving the QY by more than an order of magnitude in the high exciton density regime. Spectroscopic signatures of PL intensity and lifetime, and EEA rate constants, all showed the systematic dependence on the atomic thickness between 1L-WS$_2$ and Au plate, whose qualitative features can be accounted by exciton-exciton scattering calculations, involving a nonradiative Auger-like process, for interacting charges in 1L-WS$_2$ in proximity to a metal. Our result provides a simple scheme to regulate the fundamental process of EEA and paves the way for highly luminescent 2D light-emitting devices based on 2D-TMDs.

## Methods

**Sample preparation.** hBNs and 1L-WS$_2$ were mechanically exfoliated onto PDMS from bulk crystals[47]: WS$_2$ bulk crystals were purchased from HQ graphene and hBN bulk crystals were provided by the National Institute for Materials Science, Japan. Exfoliated hBN flakes on PDMS generally come in various thicknesses and were identified by an optical microscope and subsequently pressed and peeled off

onto ultra-flat Au substrates (Platypus Technologies) or a 300-nm-thick SiO$_2$ layer on a Si wafer at the substrate temperature of 100 °C[48]. Here, ultra-flat Au substrates were used to reduce the substrate roughness which may induce strain or incidental doping to 1L-WS$_2$. Our samples were then annealed at 150 °C in a vacuum oven for 12 h. The PL from the exfoliated 1L-WS$_2$ on PDMS was measured before the transfer. 1L-WS$_2$ samples were subsequently transferred onto the hBN/Au or hBN/SiO$_2$/Si substrates kept at 70 °C and then were annealed at 70 °C in the vacuum oven for 1-2 h. The thicknesses of hBN flakes were confirmed by atomic force microscope (XE-120, Park Systems).

**Optical measurements**. PL images and spectra were obtained with a commercial confocal microscope (Alpha-300S, WITec Instrument GmbH) equipped with a 100x objective lens (N.A.=0.9) and a frequency-doubled neodymium-doped yttrium aluminum garnet laser for 532-nm CW excitation. For TRPL measurements, the same microscope was used under pulsed excitation at 488 nm (BDL-488, Becker & Hickl GmbH), with the pulse width of 70 ps and the repetition rate of 80 MHz and an high-speed hybrid detector (HPM-100-40, Becker & Hickl GmbH) and a time-correlated single-photon counting module (TCSPC, Becker & Hickl GmbH). All measurements were conducted at room temperature.

## Data availability
All data are available within the Article and Supplementary Files, or available from the corresponding authors on reasonable request.

## Code availability
Codes used in the theoretical calculation are available from the corresponding authors on reasonable request.

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

## Acknowledgements
This work was supported by the Samsung Research Funding & Incubation Center of Samsung Electronics, under project no. SRFC-MA1802-02. A.K. acknowledges funding support from the Department of Science and Technology (DST). A. Chaves and J.D.S.F. acknowledge financial support by the Brazilian Research Council (CNPq), through the

PRONEX/FUNCAP and PQ programs. A. Chaves also acknowledges financial support by the Research Foundation - Flanders (FWO). A. Chernikov gratefully acknowledges financial support by Deutsche Forschungsgemeinschaft (DFG, German Research Foundation) through SFB 1277 (Project-ID 314695032) project B05, Emmy-Noether Grant CH 1672/1 (Project-ID: 287022282), and the Würzburg-Dresden Cluster of Excellence on Complexity and Topology in Quantum Matter ct.qmat (EXC 2147, Project-ID: 390858490). K.W. and T.T. acknowledge support from the Elemental Strategy Initiative conducted by the MEXT, Japan (Grant Number JPMXP0112101001), JSPS KAKENHI (Grant Numbers 19H05790, 20H00354 and 21H05233) and A3 Foresight by JSPS.

## Author contributions

T.L. and J.K. conceived and supervised the project, analyzed the data, and wrote the manuscript. T.T.T. and Y.K prepared the samples, performed the optical measurements. Y.L. analyzed the data and wrote the manuscript; J.D.S.F., A. Chaves, A.K., and T.L. performed theoretical calculation and wrote the manuscript. S.R. and Y.L. measured QY. T.T. and K.W. synthesized and discussed the role of high-quality hBNs. A. Chernikov and J.I.J. analyzed the data and wrote the manuscript.

## Competing interests

The authors declare no competing interests.
