## [Peer Review File · Nature Communications]

Reviewers' Comments:

Reviewer #1:

Remarks to the Author:

In the manuscript by Lee et al. "Boosting quantum yields in 2D semiconductors via proximal metal plates" the authors study a set of transition metal dichalcogenide monolayers (1L-TMDs) on different substrates. Specifically, they place 2D materials in the proximity of metal film (gold) with varying hBN spacers, and measure photoluminescence from the samples. The authors observe changes in the PL intensity, and compare the signal to a reference sample on Si substrate. The time resolved measurements of photoluminescence reveal changes in the exciton decay time, which is also pump-dependent. The findings are explained by a classical model of dipoles with different orientation.

The manuscript is well-written and represents a contribution to the broad area of 2D material physics. The experimental methodology is sound, though the theoretical description is rather weak. I am sure this is a step towards understanding the influence of substrate on PL and quantum yield, and will be of interest to specialists in the devoted subject. At the same time, I am asked to judge the potential for high impact on the broader field and, despite the claims in the introduction, do not see appeal that warrants publishing in Nature Communications. Also, several aspects in explaining results remain vague. Therefore, I cannot recommend the manuscript as an article in Nature Comm.

- One of the main points raised by authors is in enhancing the yield, aiming to improve future devices by the better substrate choice. However, for the uncorrected results the PL signal from the Si-based sample show better performance in practice. While I agree that for studying physical effects the signal should be adjusted to exclude interference, how one does it to give practical advantage? What can be done to get PL enhancement for the uncorrected case? The procedure of correction remains opaque, as it was not described in the available materials.

- The theoretical explanation behind the observed PL changes is very unclear. The authors write the Fermi golden rule for the lifetime using exciton wavefunctions Ψ_k and Ψ_{k+q} . However, to describe any nonlinear effects the expression must involve two-exciton wavefunctions. Also, for the advertised exciton-exciton annihilation study one would imagine nonlinear interaction to appear in the system Hamiltonian, but this is not present in the discussion.

- Excitons in 1L-TMD are considered as some staggered dipoles [Fig. 5a]. However, this situation does not correspond to reality (unlike for interlayer excitons in van der Waals heterostructures). The ground state in a monolayer system corresponds to 1s excitons, which on average have zero in-plane dipole moments. This is also the reason while exciton-exciton interaction is not a dipole-dipole one, but of van der Waals type ($\sim 1/d^6$). The 2D confinement also prevents excitons to get a significant out-of-plane component, and thus it is not clear how different orientation in y direction plays crucial role.

- The authors say that "optical interference effect dominates the observed PL dependence on the hBN thickness." [lines 123-124]. This goes in contrast to the main claim of the manuscript, where the study of exciton-exciton interaction is highlighted as a major effect behind quantum yield enhancement. Overall, it is difficult to distinguish the two contribution. The conclusion in line 131 about thus potentially misleads readers.

To conclude, this is an overall good study, just not at the level of required for Nature Comm.

Reviewer #2:

Remarks to the Author:

In their manuscript „Boosting quantum yields in 2D semiconductors via proximal metal plates" the authors use time-resolved optical measurements to demonstrate that nonradiative exciton decay channels in monolayer WS₂ can be suppressed by an ultra-thin Au substrate separated by multilayers of hexagonal boron nitride (hBN). This effect is explained by efficient screening of

dipolar exciton-exciton annihilation (EEA) processes that reduce the quantum yield of 2D light emitters at elevated excitation densities. To corroborate this conclusion, the authors compare the measured dependence of EEA rates on hBN layer thickness to results from a semiclassical screening model.

From a technological perspective, controlling nonradiative decay channels is very interesting to increase the performance of potential 2D light-emitting devices. As discussed in the manuscript, Auger interaction is often considered the most fundamental limitation of quantum yield, since it is an intrinsic effect. Therefore, in my opinion, the presented results are very promising. There is a lot of valuable insight into different aspects such as enhancement of emission due to optical interference and dependence of the EEA rate on substrate materials and thickness.

I would, however, not say that the manuscript provides fundamental insight into the physics of exciton-exciton annihilation. The used semiclassical screening model is based on very rough assumptions, which I will address in more detail below. Moreover, the central theory-experiment comparison in Figure 5c is, in my eyes, not as affirmative as the authors suggest: The absolute scale of the EEA rates is adjusted by a scaling factor and the trend with hBN thickness could as well be given by a flat curve within the experimental error bars. I do acknowledge that a microscopic theory of EEA is not yet available, but the presented model should not be oversold.

In summary, the authors provide valuable experimental data with strong implications for optoelectronics applications, while I am not so much convinced of the semiclassical model. Before I can recommend publication of the manuscript in Nature Communications, I would like to ask the authors to address my questions and comments below.

By the way, I could not find a Supplementary Information file that is mentioned several times in the manuscript. Therefore, it is not clear to me what parameters are used to describe the dielectric properties of the substrate layers.

1) The authors cite Ref. [20] in the context of exciton-exciton interaction, which is a bit misleading since Ref. [20] is concerned with bandgaps and exciton binding energies.

2) On page 7, electrostatic depletion of electrons is explained with the WS₂ conduction band edge being higher than the Fermi energy of gold, which are 4 eV and 5.2 eV, respectively. In what sense is 4 eV higher than 5.2 eV?

4) Are the dielectric properties of Au described by a constant epsilon, or is the metallic screening taken into account?

5) On page 11, Ref. [40] is cited for the transfer-matrix scheme used to fix the boundary conditions for Poisson's equation. The authors could acknowledge here another publication that brought forward a similar approach at around the same time: [Florian et al., Nano Lett. 2018, 18, 2725]

6) The authors model the EEA as a dipole-dipole interaction, which is certainly an approximation since exchange terms are neglected. [M. Combescot and O. Betbeder-Matibet 2002 EPL 58 87] I think this should be declared.

7) The semiclassical model used by the authors leads to the peculiar effect that screening by the dielectric environment depends on the actual exciton density, since the latter fixes the distance between two excitons. From a microscopic perspective, I would expect the exciton density to enter the EEA rate only via exciton distribution functions (approximately Bose functions). The matrix elements that describe the efficiency of exciton-exciton interaction should not depend on density unless screening due to the excitons themselves is taken into account. In my opinion, it is problematic to describe a gas of interacting excitons by classical particles at fixed positions instead of quantum-mechanical distribution functions. Let me try to clarify this:

What the authors might have in mind is an analogy to the exciton problem, where electron and hole are approximately separated by the Bohr radius. The separation determines how the Coulomb

field lines penetrate the dielectric environment. Then, the screening efficiency depends on the average electron-hole distance. This is microscopically encoded in the „sampling“ of the momentum-dependence of the dielectric function by the exciton wave function via the Wannier equation. In this situation, it is legitimate in my eyes to think of the exciton as two particles at a certain (average) distance. However, this situation involves no exciton density. It rather assumes that only two carriers are present, corresponding to zero exciton density. The result are the eigenenergies and -states of the given many-body system, which can then be occupied by a certain density of excitons. In a similar way, the eigenstates of two interacting excitons could be determined, yielding bound states (biexcitons) and scattering states, which again do not know about an exciton density.

The EEA is now given by the dynamics of a density of excitons taking place in terms of eigenstates that are determined by the zero-density problem in the presence of a dielectric environment. Again, it is the exciton eigenstates that enter the exciton-exciton matrix elements, but not the exciton density.

I would be interested to hear the authors' opinion about this.

8) A point that may be connected to 7): The form of Eq. (4) is questionable. What are the states Ψ that enter Fermi's golden rule? Is one of them a two-exciton state, while the other one contains one exciton? What are the momenta k and q in this context? In particular, q does not appear on the LHS of the equation.

Also, how can one reasonably estimate that the spatial dependence of V is negligible with respect to the wave functions? I have the impression that real-space and momentum representations are mixed up here in a strange way. Maybe there is more justification for this in the SI.

A final remark on points 7) and 8): I am convinced that the dielectric environment does influence the efficiency of EEA, but I doubt that the used theory correctly captures the screening dependence.

Editor's/Reviewers' comments:

We deeply appreciate all the reviewers for reviewing our manuscript and commenting valuable suggestions and advice to strengthen our manuscript. We have revised our manuscript according to the reviewer's comments and addressed the reviewer's comments point-by-point in this response letter. The changes made in the manuscript are written in red.

Reviewer: 1

Comments:

In the manuscript by Lee et al. "Boosting quantum yields in 2D semiconductors via proximal metal plates" the authors study a set of transition metal dichalcogenide monolayers (1L-TMDs) on different substrates. Specifically, they place 2D materials in the proximity of metal film (gold) with varying hBN spacers, and measure photoluminescence from the samples. The authors observe changes in the PL intensity and compare the signal to a reference sample on Si substrate. The time resolved measurements of photoluminescence reveal changes in the exciton decay time, which is also pump-dependent. The findings are explained by a classical model of dipoles with different orientation.

The manuscript is well-written and represents a contribution to the broad area of 2D material physics. The experimental methodology is sound, though the theoretical description is rather weak. I am sure this is a step towards understanding the influence of substrate on PL and quantum yield, and will be of interest to specialists in the devoted subject. At the same time, I am asked to judge the potential for high impact on the broader field and, despite the claims in the introduction, do not see appeal that warrants publishing in Nature Communications. Also, several aspects in explaining results remain vague. Therefore, I cannot recommend the manuscript as an article in Nature Comm.

Reply: The reviewer has seen the main points of our manuscript. We are thankful for the positive assessment and useful criticism. We carried out additional experiments to emphasize the high impact of our results on the area of 2D materials research. We also extensively revised our theoretical model.

1. One of the main points raised by authors is in enhancing the yield, aiming to improve future devices by the better substrate choice. However, for the uncorrected results the PL signal from the Si-based sample show better performance in practice. While I agree that for

studying physical effects the signal should be adjusted to exclude interference, how one does it to give practical advantage? What can be done to get PL enhancement for the uncorrected case? The procedure of correction remains opaque, as it was not described in the available materials.

Reply: We would like to thank the reviewer for this insightful comment. We agree that in the original manuscript the advantage of a higher QY by a metal substrate was not clearly shown because of the heavy influence of the substrate interference on the observed PL intensity. However, we point out that the measured PL intensity (without the adjustment on the substrate interference effect) of 1L-WS₂ on 7 nm hBN/Au was already 2.5 times higher than that on the SiO₂/Si substrate, although the enhancement factor (EF) was 3 times less, as shown in Fig. 1(d) and 1(e). At the high exciton density, the measured PL from 1L-WS₂ on 7 nm hBN/Au was 10 times higher than that on the SiO₂/Si substrate. This clearly demonstrated the practical advantages of our substrate configuration even without the interference adjustment. We have modified the original text as follow to make this point clear.

“At first glance, it seems that the relative PL intensity from 1L-WS₂ on 7 nm hBN/Au is 2.5 times higher than that from 1L-WS₂ directly on the SiO₂/Si substrate (Figs. 1d and 1e). However, we confirmed that the actual QY of the former is about five times larger since the SiO₂/Si substrate gains the PL brightness via the EF by more constructive inference of multiple internal reflections, as denoted by the star on the right vertical axis in Fig. 1e.”

In addition, in order to directly demonstrate the effect of metal proximity, we have prepared a new set of 1L-WS₂ on two different hBN thicknesses of 5 nm and 34 nm on Au substrates. In this case, we have encapsulated the 1L-WS₂ with top hBN with 52 nm thickness. (See the schematics in Fig. R1) Our calculation indicated that the ratio of the EF between 34 nm and 5 nm hBN cases can be reduced to 1.25, which was originally 10.3, by employing the top hBN, thereby indicating that hBN encapsulation can nearly remove the complicated interference effect. In addition, we have newly added differential reflectance spectra to confirm stronger absorption for thicker hBN on Au. In the revision, we have added the following text.

“Differential reflection spectra of 1L-WS₂ also confirmed stronger absorption at thicker hBN on the Au substrate and we showed that the effect of hBN thickness on the measured PL intensity can be nearly flatten by encapsulating 1L-WS₂ with thick (~50 nm) hBN (Figs. S2

and S3 in the Supplementary Information).”

Fig. R1. (Fig. S2 in the revised Supplementary Information) Differential reflectance spectra obtained from 1L-WS₂ on different thicknesses of hBN on Au. Increasing absorption with increasing hBN thickness is clearly shown.

In Fig. R2, we show the measured PL spectra of 1L-WS₂ on 5 nm and 34 nm hBN on Au with (upper panels) and without (lower panels) top hBN obtained at 6 W/cm² and 15 kW/cm². These PL spectra were obtained by averaging the whole area of 1L-WS₂ of the corresponding bottom hBN thickness (indicated by the dashed lines in the inset, showing PL intensity images of the sample). With top hBN, at 6 W/cm² (or $7 \times 10^{16} \text{ cm}^{-1} \text{ s}^{-1}$) where EEA has no effect on the PL intensity, 1L-WS₂ on 5 nm hBN showed a 30% lower PL intensity compared to 34 nm hBN, consistent with the EF. At 15 kW/cm² ($4 \times 10^{20} \text{ cm}^{-1} \text{ s}^{-1}$) where EEA is strongly active, 1L-WS₂ on 5 nm hBN/Au showed a 25% higher PL intensity than 34 nm hBN, directly showing the effect of metal screening by thinner hBN on the Au substrate.

We would like to also mention that in many practical applications of 1L-TMD as photonic devices, hBN encapsulation is frequently used for the purpose of preserving the quality of 1L-TMDs. Therefore, the superior luminescence (without the adjustment on the interference effect) due to the suppressed EEA shown in our hBN-encapsulated samples indeed implies

the practical advances of our innovative scheme.

Fig. R2. (Fig. S3 in the revised Supplementary Information) Observed PL spectra and intensity images of 1L-WS₂ on hBN/Au with (upper panels) and without (lower panels) top 52 nm hBN, obtained at low (left panels) and high (right panels) excitation powers. Insets: PL intensity images of 1L-WS₂ on 5 nm hBN and 34 nm hBN on the Au substrate. Contrast is normalized for each image.)

2. The theoretical explanation behind the observed PL changes is very unclear. The authors write the Fermi golden rule for the lifetime using exciton wavefunctions Ψ_k and Ψ_{k+q} . However, to describe any nonlinear effects the expression must involve two-exciton wavefunctions. Also, for the advertised exciton-exciton annihilation study one would imagine

nonlinear interaction to appear in the system Hamiltonian, but this is not present in the discussion.

Reply: We agree that the theoretical explanation in the previous version was unclear, therefore, we have made strong modifications to the theoretical model used to support the experimental results. This is now explained in details in the Supplementary Information of the resubmitted version. In summary, we have developed a model for nonradiative EEA assuming an Auger process where one exciton recombines, while the energy released by such recombination is used to unbind the other exciton, leaving a high-energy unbound electron-hole pair after the process. Under some approximations, the model is used to explain the dependence of the EEA rate on the hBN thickness. Moreover, we have also improved the exciton-exciton interaction potential calculations proposed in the previous version of the paper, taking into account the exciton wave functions in a more appropriate manner.

3. Excitons in 1L-TMD are considered as some staggered dipoles [Fig. 5a]. However, this situation does not correspond to reality (unlike for interlayer excitons in van der Waals heterostructures). The ground state in a monolayer system corresponds to 1s excitons, which on average have zero in-plane dipole moments. This is also the reason while exciton-exciton interaction is not a dipole-dipole one, but of van der Waals type ($\sim 1/d^6$). The 2D confinement also prevents excitons to get a significant out-of-plane component, and thus it is not clear how different orientation in y direction plays crucial role.

Reply: The referee is correct: exciton-exciton interactions are indeed of van der Waals type. Actually, the way we presented our theoretical model for exciton-exciton interactions in the previous version of the paper was a bit misleading – we started with dipole-dipole interactions, which is the actual interaction between the two electrons and two holes involved in the two excitons. In fact, this is also how the calculations that lead to van der Waals interactions begin. However, from this point onwards, we took an approximation that is possibly inaccurate: we assumed the electron-hole separation in each exciton to be roughly the Bohr radius and performed an average in the angular coordinates. Van der Waals interactions, on the other hand, are obtained by assuming such dipole-dipole interactions as the potential in Schrodinger equation and solving it e.g. by perturbation methods. In second-order perturbation, the $1/R^3$ dipole-dipole interaction ends up as the $1/R^6$ interaction observed in the van der Waals potential (with R as the exciton-exciton separation). Since we verified that our method did not lead to the correct power of R in the

interaction potential even in the case where we assumed Coulomb interaction, we decided to improve our exciton-exciton calculations.

Firstly, notice that one should not expect a perfect van der Waals potential in the case we investigate here, since the $1/R^6$ interaction is ultimately a result of Coulomb interactions, which is not the case for electron-hole interactions in the 2D systems investigated here, where non-uniform dielectric screening plays an important role. The presence of the metal changes even more the form of the potential, which forces us to re-derive exciton-exciton interactions for such unconventional Coulomb potentials.

In a naive approximation, one can still re-develop the calculations that lead to van der Waals interactions starting from the screened Coulomb interaction potential between charges in a strictly 2D plane, separated from the metal by a distance d . The metal would simply induce image charges that modify the Coulomb interactions by adding terms proportional to $1/\sqrt{R^2 + 4d^2}$. It is straightforward to demonstrate that in this case, the effective van der Waals interactions acquire the form $\sim A*[1/R^3 + 1/\sqrt{R^2 + 4d^2}]^2$, where the constant A carries information about the Bohr radius of the excitons, which in turn depends on the hBN thickness. This result clearly converges to the usual van der Waals interaction as $d \rightarrow \text{infinity}$, when the metal is not supposed to play a role, as expected. However, results in this approximation are too sensitive to the distance “ d ” (i.e. hBN thickness), which suggests that this calculation does not capture the extra screening due to the hBN spacer properly. Indeed, the limit $d \rightarrow \text{infinity}$ in this case leads to Coulomb interactions, whereas it is well known that interactions in such a 2D slab are rather of the Rytova-Keldysh type.

Therefore, in order to properly account for the screening of the 2D WS_2 slab itself, the hBN spacer and the metal, we have performed a full numerical calculation of the exciton-exciton interaction in the same fashion as in the calculations for van der Waals interactions, but now with the interaction potential obtained by solving Poisson equation for the vacuum- WS_2 -hBN-Au system. This method and its results are now detailed in the Supplementary Information of the resubmitted version. Interestingly, the exciton-exciton interactions calculated with this method depend on the hBN thickness in a similar way as the EEA rate in the experiments, and therefore, this conclusion is still maintained in the paper.

4. The authors say that “optical interference effect dominates the observed PL dependence on the hBN thickness.” [lines 123-124]. This goes in contrast to the main claim of the manuscript, where the study of exciton-exciton interaction is highlighted as a major effect behind quantum yield enhancement. Overall, it is difficult to distinguish the two contribution.

The conclusion in line 131 about this potentially misleads readers.

Reply: We apologize for the confusion that we caused by not stating that PL image and spectra shown in Fig. 1 were obtained with a very low laser power of only 2 nW, in which EEA is quite negligible. We have added the following text to clearly indicate the excitation condition.

“We used the low laser power of 2 nW to assess the QY without the effect of EEA.”

Furthermore, we have modified the texts to describe that the observed PL intensity of 1L-WS₂ on 7 nm hBN/Au is 2.5 times higher than that on SiO₂/Si (Fig. 1d and 1e) and, if the substrate interference is considered, the QY of the former is about 5 times higher.

“At first glance, it seems that the relative PL intensity from 1L-WS₂ on 7 nm hBN/Au is 2.5 times higher than that from 1L-WS₂ directly on the SiO₂/Si substrate (Figs. 1d and 1e). However, we confirmed that the actual QY of the former is about five times larger since the SiO₂/Si substrate gains the PL brightness via the EF by more constructive interference of multiple internal reflections, as denoted by the star on the right vertical axis in Fig. 1e.”

Thus, we believe that the statement that “*..the proximity to the Au surface increased the QY of 1L-WS₂.*” is valid. Furthermore, in order to directly show (without the interference effect) the effect of metal proximity suppressing EEA and boosting the QY at the high exciton density, we have newly prepared 1L-WS₂ samples covered with 55-nm-thick hBN and with two different bottom hBN thicknesses of 5 nm and 34 nm (Fig. R2). In this configuration, the calculated EF due to the substrate interference was only 1.25 times higher for the sample on 34 nm bottom hBN than on 5 nm hBN/Au and the observed PL intensity of the encapsulated 1L-WS₂ was higher at thinner (5 nm) bottom hBN than thicker (34 nm) hBN on Au in the high exciton density, directly proving that the suppression of EEA was more effective for thinner hBN on Au.

To conclude, this is an overall good study, just not at the level of required for Nature Comm.

Reply: We truly thank the reviewer for the advice. We hope our revision with new experimental results and the expanded theoretical model has convinced the reviewer of the significance and the originality of our work.

Reviewer: 2

Comments:

In their manuscript “Boosting quantum yields in 2D semiconductors via proximal metal plates“ the authors use time-resolved optical measurements to demonstrate that nonradiative exciton decay channels in monolayer WS₂ can be suppressed by an ultra-thin Au substrate separated by multilayers of hexagonal boron nitride (hBN). This effect is explained by efficient screening of dipolar exciton-exciton annihilation (EEA) processes that reduce the quantum yield of 2D light emitters at elevated excitation densities. To corroborate this conclusion, the authors compare the measured dependence of EEA rates on hBN layer thickness to results from a semiclassical screening model.

From a technological perspective, controlling nonradiative decay channels is very interesting to increase the performance of potential 2D light-emitting devices. As discussed in the manuscript, Auger interaction is often considered the most fundamental limitation of quantum yield, since it is an intrinsic effect. Therefore, in my opinion, the presented results are very promising. There is a lot of valuable insight into different aspects such as enhancement of emission due to optical interference and dependence of the EEA rate on substrate materials and thickness.

I would, however, not say that the manuscript provides fundamental insight into the physics of exciton-exciton annihilation. The used semiclassical screening model is based on very rough assumptions, which I will address in more detail below. Moreover, the central theory-experiment comparison in Figure 5c is, in my eyes, not as affirmative as the authors suggest: The absolute scale of the EEA rates is adjusted by a scaling factor and the trend with hBN thickness could as well be given by a flat curve within the experimental error bars. I do acknowledge that a microscopic theory of EEA is not yet available, but the presented model should not be oversold.

Reply: We thank the referee for the positive view of our work. We agree that the theoretical model should not be oversold. We also understand the model in the previous version of the manuscript was lacking of deeper physical insight, and therefore, in this resubmitted version, we substantially improved the theoretical model. Nevertheless, as the referee points out, it is very hard to develop a rigorous microscopic theory of EEA and obtain accurate predictions of annihilation rates. Since we could only obtain trends that are compared to the experimental data through a scaling factor, we have decided to move most of the theory to the Supplementary Information.

In summary, the authors provide valuable experimental data with strong implications for optoelectronics applications, while I am not so much convinced of the semiclassical model. Before I can recommend publication of the manuscript in Nature Communications, I would like to ask the authors to address my questions and comments below.

By the way, I could not find a Supplementary Information file that is mentioned several times in the manuscript. Therefore, it is not clear to me what parameters are used to describe the dielectric properties of the substrate layers.

Reply: We apologize the exclusion of Supplementary Information that occurred through a technical glitch. Please find the revised Supplementary Information that includes the details on the calculation of the interference contribution to the observed PL.

1. The authors cite Ref. [20] in the context of exciton-exciton interaction, which is a bit misleading since Ref. [20] is concerned with bandgaps and exciton binding energies.

Reply: We appreciate the reviewer for the careful reading. We agree on the reviewer's opinion and have removed the sentence and the reference #20.

2. On page 7, electrostatic depletion of electrons is explained with the WS₂ conduction band edge being higher than the Fermi energy of gold, which are 4 eV and 5.2 eV, respectively. In what sense is 4 eV higher than 5.2 eV?

Reply: We are sorry for the confusion. We have added the minus (-) sign in front of these energy values to clarify that the quoted values are measured from the vacuum level.

3. Are the dielectric properties of Au described by a constant epsilon, or is the metallic screening taken into account?

Reply: We did not use a constant epsilon for Au. Metallic screening was taken into account in the model. This is now clarified in the text.

4. On page 11, Ref. [40] is cited for the transfer-matrix scheme used to fix the boundary conditions for Poisson's equation. The authors could acknowledge here another publication that brought forward a similar approach at around the same time: [Florian et al., Nano Lett. 2018, 18, 2725]

Reply: We thank the referee for the suggestion. This reference has been added to the resubmitted version of the manuscript.

5. The authors model the EEA as a dipole-dipole interaction, which is certainly an approximation since exchange terms are neglected. [M. Combescot and O. Betbeder-Matibet 2002 EPL 58 87] I think this should be declared.

Reply: Indeed, we neglected exchange terms in our calculation, as an approximation. This is now clarified in the resubmitted version, along with the reference mentioned by the referee.

6. The semiclassical model used by the authors leads to the peculiar effect that screening by the dielectric environment depends on the actual exciton density, since the latter fixes the distance between two excitons.

Reply: The referee summarizes our approach correctly: we approximate that the exciton-exciton separation R is roughly controlled by the exciton density. However, the screening by the dielectric environment in our model does not actually depend on the density. We calculate the interaction potential between pairs of charges, accounting for the dielectric environment (similar to the Rytova-Keldysh potential, but accounting also for the presence of a metal), and then we construct exciton-exciton interaction potentials using these pairwise interactions. Afterwards, we estimate the exciton-exciton separation as being, on average, $R \sim \sqrt{1/n}$, where n is the exciton density, and then we are able to estimate the average exciton-exciton interaction energy in the system, as being the value of the exciton-exciton interaction potential at R . However, as we will clarify later in this reply letter, in this resubmitted version, we have not only improved the exciton-exciton interaction potential calculations, but also provided a proper estimate of the EEA rate that depends on the hBN thickness, by modelling EEA as an Auger process.

From a microscopic perspective, I would expect the exciton density to enter the EEA rate only via exciton distribution functions (approximately Bose functions). The matrix elements that describe the efficiency of exciton-exciton interaction should not depend on density unless screening due to the excitons themselves is taken into account. In my opinion, it is problematic to describe a gas of interacting excitons by classical particles at fixed positions instead of quantum-mechanical distribution functions. Let me try to clarify this:

Reply: We fully agree with all the comments made by the referee. Notice that in the previous version of the paper, we discussed EEA as being an indirect measure of exciton-exciton

interactions. What we calculated there was the effect of the metal and hBN spacer on exciton-exciton interactions, and then we suggested that these interactions for a fixed exciton-exciton distance (i.e. a fixed density, as in the experimental data for EEA in Fig. 5c) exhibit a similar trend as compared to the EEA rates. But we understand that even this calculation was still oversimplified by this semi-classical approach, therefore, in this new version, we have improved the exciton-exciton interaction calculations by using the pairwise interaction potentials discussed above in the Schrodinger equation for two excitons and obtaining the energy of the pair of excitons via variational principle. Our results converge to the van der Waals interaction between excitons in the limit of Coulomb interactions between electrons and holes and in the absence of a metal, as expected. We have concluded that the exciton-exciton interaction and the EEA rate seem to have similar trends as the hBN thickness increases.

However, in the resubmitted manuscript, we have also included an approximate calculation for the EEA rates (rather than exciton-exciton interactions). In this calculation, indeed, the density enters only in the distribution functions, not in the actual matrix elements.

What the authors might have in mind is an analogy to the exciton problem, where electron and hole are approximately separated by the Bohr radius. The separation determines how the Coulomb field lines penetrate the dielectric environment. Then, the screening efficiency depends on the average electron-hole distance. This is microscopically encoded in the „sampling“ of the momentum-dependence of the dielectric function by the exciton wave function via the Wannier equation. In this situation, it is legitimate in my eyes to think of the exciton as two particles at a certain (average) distance. However, this situation involves no exciton density. It rather assumes that only two carriers are present, corresponding to zero exciton density. The result are the eigenenergies and -states of the given many-body system, which can then be occupied by a certain density of excitons. In a similar way, the eigenstates of two interacting excitons could be determined, yielding bound states

(biexcitons) and scattering states, which again do not know about an exciton density.

The EEA is now given by the dynamics of a density of excitons taking place in terms of eigenstates that are determined by the zero-density problem in the presence of a dielectric environment. Again, it is the exciton eigenstates that enter the exciton-exciton matrix elements, but not the exciton density.

I would be interested to hear the authors' opinion about this.

Reply: We thank the referee for emphasizing the difference between our previous discussion involving exciton-exciton interactions and a proper EEA rate calculation. This has inspired us to improve the discussion about EEA by modeling a nonradiative scattering problem between two excitons. Many possibilities can lead to nonradiative processes, but most of them would carry on the dielectric environment encoded in the interaction potential between charges. As a sample case, we have chosen the situation where the initial state is composed of two excitons, and the interaction between the two excitons leads to a final state composed by a single, high-energy, unbound electron-hole pair. In this Auger process, the energy from the recombination of one exciton is transferred to the other exciton, resulting in such an unbound electron-hole pair. Even for this case, an actual calculation of the EEA rate as a function of hBN thickness is a very challenging task, therefore, after some approximations we could only demonstrate that an Auger process such as described here also follows approximately the same trend as the experimentally observed EEA as the hBN thickness increases.

7. A point that may be connected to 7): The form of Eq. (4) is questionable. What are the states Ψ that enter Fermi's golden rule? Is one of them a two-exciton state, while the other one contains one exciton? What are the momenta k and q in this context? In particular, q does not appear on the LHS of the equation.

Also, how can one reasonably estimate that the spatial dependence of V is negligible with respect to the wave functions? I have the impression that real-space and momentum representations are mixed up here in a strange way. Maybe there is more justification for this in the SI.

Reply: In the resubmitted version of the manuscript, the two approaches, namely, the one involving (i) exciton-exciton interactions, and the other (ii) where EEA is interpreted as an Auger process, are now clearly separated, which solves the issue. Now, Eq. (4) and the appropriate Fermi's golden rule only enter the discussion, where the initial and final wave functions are better clarified.

A final remark on points 7) and 8): I am convinced that the dielectric environment does influence the efficiency of EEA, but I doubt that the used theory correctly captures the screening dependence.

Reply: We understand the concerns of the referee, and therefore, as discussed in the previous replies, we have modified our theoretical model accordingly. We thank the referee for all comments and suggestions, which have helped us to develop a much deeper understanding of the EEA process.

Reviewers' Comments:

Reviewer #1:

Remarks to the Author:

In the revised version of the manuscript Lee et al. improved the explanation of experimental results, now supported by additional measurements. The authors also considered seriously concerns from the previous revision, and made efforts to improve theory. Thus, I am now in favour of publication of the manuscript in Nature Communications.

The theory is certainly improved, but yet cannot be used to give quantitative predictions. I suggest that the authors tone it down, and do not rely too much on the theoretical description.

Also, the Auger process as stated has final and initial states interchanged, and does not conserve charge and spin (transition is between two excitons and one e-h pair, although it should be equal number of carriers on both sides). Can the authors look up for a description that is more formal? With these corrections the manuscript shall be in a decent shape.

Reviewer #2:

Remarks to the Author:

In their revised manuscript, the Authors addressed most of my questions and comments adequately. In particular, the theoretical foundations of the discussion of EEA efficiency depending on the dielectric environment are much more solid now.

It only remains unclear to me what parameters are used to describe the dielectric properties of the different materials. The Authors provide a table S1 that contains the refractive indices, but not the dielectric constants that enter equation (2) via transfer matrices. In my review, I asked if the dielectric properties of Au are described by metallic screening or by a constant epsilon. The Authors declare that metallic screening is used, but I can not find any more information about how this is done. Is the characteristic momentum dependence of metallic screening taken into account, which is fundamentally different from screening in an insulating material?

Moreover, now that I could read the Supplementary Information, I have an additional remark. I am afraid that equations (S5) are not correct. A solution of Maxwell's equations for a thin film assuming a simple Lorentzian absorbance, e.g. using transfer matrices, yields the relation $\Delta R / R = (n_{\text{sub}}+1)/(n_{\text{sub}}-1)*A$ in the limit of small A (and a similar relation for $\Delta T / T$). This is consistent with the discussion in [arXiv:1801.00402], in particular equations (14)-(16).

Editor's/Reviewers' comments:

We deeply appreciate all the reviewers for reviewing our manuscript and commenting valuable suggestions and advice to strengthen our manuscript. We have revised our manuscript according to the reviewer's comments and addressed the reviewer's comments point-by-point in this response letter.

Reviewer: 1

Comments:

In the revised version of the manuscript Lee et al. improved the explanation of experimental results, now supported by additional measurements. The authors also considered seriously concerns from the previous revision, and made efforts to improve theory. Thus, I am now in favour of publication of the manuscript in Nature Communications.

The theory is certainly improved, but yet cannot be used to give quantitative predictions. I suggest that the authors tone it down, and do not rely too much on the theoretical description.

Reply: We are grateful for the reviewer's positive remarks and suggestions. We made a few changes on wordings in abstract, main text (p15) and the caption of Fig. 5, so that the theoretical results are presented as a plausible explanation of the experiments rather than sounding too conclusive.

Also, the Auger process as stated has final and initial states interchanged, and does not conserve charge and spin (transition is between two excitons and one e-h pair, although it should be equal number of carriers on both sides). Can the authors look up for a description that is more formal? With these corrections the manuscript shall be in a decent shape.

Reply: We thank the referee for the interesting comment. It is worth noting that electron-hole exchange interaction, which governs the fine structure of the excitons (e.g. spin-singlet/triplets splittings), is not included in this study since it is hardly relevant to understanding the key phenomena of the experiments. Hence, keeping to a minimal notation convention, the initial state we considered consists of two excitons, as represented by $|i\rangle = |X_1\rangle|X_2\rangle$ in the Supplementary Material, where excitons 1 and 2 are given by wave functions $|X_1\rangle$ and $|X_2\rangle$, respectively. The scattering process assume one of the excitons recombines, and its resulting excess energy is used to unbind the counterpart exciton, therefore, the final wave function is just an unbound electron-hole pair, represented by $|f\rangle = |\vec{k}_e\vec{k}_h\rangle$. This is the exact same notation as in [Phys. Rev. B 54, 16625 (1996), Phys

Rev B 73, 245424 (2006)], references [42, 43] in our main manuscript, where similar processes are described. Also, charge and spin *are* conserved in the process: the electron and hole of the first exciton just recombine, but they were not “removed” from the physical picture. The system has a neutral initial state (two electrons + two holes) and ends up with a neutral final state (one electron + one hole). Such exciton-exciton Auger process has been described in the several previous papers in the literature that we cited – we apologize for the misunderstanding and, in order to make it more clear, we now added clear statements about the initial and final function as well as references to the previous papers where this topic is discussed in the literature.

Reviewer: 2

Comments:

In their revised manuscript, the Authors addressed most of my questions and comments adequately. In particular, the theoretical foundations of the discussion of EEA efficiency depending on the dielectric environment are much more solid now.

It only remains unclear to me what parameters are used to describe the dielectric properties of the different materials. The Authors provide a table S1 that contains the refractive indices, but not the dielectric constants that enter equation (2) via transfer matrices. In my review, I asked if the dielectric properties of Au are described by metallic screening or by a constant epsilon. The Authors declare that metallic screening is used, but I can not find any more information about how this is done. Is the characteristic momentum dependence of metallic screening taken into account, which is fundamentally different from screening in an insulating material?

Reply: We thank for the reviewer's positive remarks. We apologize for the lack of more precise information on the method we used for Eq. (2). Indeed, what we mean by "metallic screening" does not account for its characteristic momentum dependence – we account for the metal screening simply by assuming the metal as an interface at which, as a boundary condition, the electrostatic potential is set to zero in the solution for Poisson equation that leads to the electrostatic potential generated by a charge in the system. As one solves Poisson equation for a charge in a bulk material, one obtains the Coulomb interaction potential. Instead, in a slab in vacuum, Rytova [N. S. Rytova, Proc. MSU, Phys., Astron. 3, 30 (1967)] and Keldysh [L. V. Keldysh, JETP Lett. 29, 658 (1979)] demonstrated that the charge-charge interaction potential exhibits a different profile, which has been indirectly verified e.g. by numerous experiments showing a non-hydrogenic (namely, non-Coulomb-like electron-hole interaction) excitonic Rydberg series. What we did in the present manuscript is to follow the steps of Rytova-Keldysh, but now in a system consisting of a slab separated from Au by hBN. In fact, we verified the validity of all the expected limits in our solution for the interaction potential: (i) if the slab is infinitely thick, we obtain the Coulomb interaction (constant epsilon); (ii) if the slab is infinitely thin, we obtain the interaction between electrons and holes and their image charges inside the metal; (iii) if the hBN spacer is infinitely thick, we obtain the Rytova-Keldysh potential. These correct limits help us to ensure the interaction potential we consider in Eq. (2) is indeed a proper solution to the electron-hole interaction problem. This is now clarified in the manuscript.

Moreover, now that I could read the Supplementary Information, I have an additional remark. I am afraid that equations (S5) are not correct. A solution of Maxwell's equations for a thin film assuming a simple Lorentzian absorbance, e.g. using transfer matrices, yields the relation $\Delta R / R = (n_{\text{sub}}+1)/(n_{\text{sub}}-1)*A$ in the limit of small A (and a similar relation for $\Delta T / T$). This is consistent with the discussion in [arXiv:1801.00402], in particular equations (14)-(16).

Reply: We appreciate the reviewer for pointing out the mistake that actually corresponds to the case when the sample is free standing. We also noticed that λ^2 was omitted too in Eq. S5, which now has been corrected. Moreover, we have included Eq. S6 that properly takes into account the absorption (A) of the samples on quartz substrates. We have found that this correction slightly reduces A by a factor of 0.6. However, we emphasize that this change does not affect the estimation of the a-QY of our TMD samples because it was compared with the reference (R6G-embedded PMMA thin film with the known QY) on the quartz substrate. In other words, the same ration of reduction in A value for the TMD and the reference canceled each other in estimating the a-QY.